# Influence of Fraction Particle Size of Pure Straw and Blends of Straw with Calcium Carbonate or Cassava Starch on Pelletising Process and Pellet

**DOI:** 10.3390/ma13204623

**Published:** 2020-10-16

**Authors:** Aleksander Lisowski, Patryk Matkowski, Leszek Mieszkalski, Remigiusz Mruk, Mateusz Stasiak, Michał Piątek, Adam Świętochowski, Magdalena Dąbrowska, Paweł Obstawski, Tomasz Bakoń, Krzysztof Karpio

**Affiliations:** 1Department of Biosystems Engineering, Institute of Mechanical Engineering, Warsaw University of Life Sciences, Nowoursynowska 166, 02-787 Warsaw, Poland; patryk_matkowski@sggw.edu.pl (P.M.); michal_piatek@sggw.edu.pl (M.P.); adam_swietochowski@sggw.edu.pl (A.Ś.); magdalena_dabrowska@sggw.edu.pl (M.D.); 2Department of Production Engineering, Institute of Mechanical Engineering, Warsaw University of Life Sciences, Nowoursynowska 166, 02-787 Warsaw, Poland; leszek_mieszkalski@sggw.edu.pl (L.M.); remigiusz_mruk@sggw.edu.pl (R.M.); 3Institute of Agrophysics, Polish Academy of Sciences, Doświadczalna 4, 20-290 Lublin, Poland; m.stasiak@ipan.lublin.pl; 4Department of Fundamentals of Engineering and Energy, Institute of Mechanical Engineering, Warsaw University of Life Sciences, Nowoursynowska 166, 02-787 Warsaw, Poland; pawel_obstawski@sggw.edu.pl (P.O.); tomasz_bakon@sggw.edu.pl (T.B.); 5Department of Applied Mathematics, Institute of Information Technology, Warsaw University of Life Sciences, Nowoursynowska 166, 02-787 Warsaw, Poland; krzysztof_karpio@sggw.edu.pl

**Keywords:** biomass, additives, pressure agglomeration, pellet strength, water absorption

## Abstract

The aim of this study was to investigate the pressure agglomeration process of wheat straw (WS) and the blends of WS with calcium carbonate (CC) or cassava straw (CS) with a ratio of 6% wt./wt. from seven separate fractions with sizes in the range of 0.21–2.81 mm. The agglomeration was performed at a moisture of 30% wb and a material temperature of 78 °C, with a dose of 0.1 g, in a die of diameter 8 mm and height 80 mm. The effects of the process were evaluated based on the compaction parameters and the pellets’ density, tensile strength, and water absorption. The incorporation of additives into the WS improved the pellet process and quality. Refined results were achieved after adding CC, as compared to those achieved after adding CS, and the preferred particle size was in the range of 1.00–1.94 mm. This was because, under the given conditions, the back pressure in the die chamber significantly increased, allowing the achievement of a single pellet density of 800 kg·m^−3^. The pellets were resistant to compressive loads and cracked only at tensile strength of 6 MPa and a specific compression work of 6.5 mJ·mm^−2^. The addition of CC to the WS improved the strength of the adhesive and the cohesive bonds between the particles. The water absorption for the uncrushed pellets was considerably less than that for crushed pellets, which results in the safer storage of uncrushed pellets and excellent moisture absorption of crushed pellets. The addition of CC to the WS offers benefits in the form of pellet strength with a high water absorption capability. Notably, a study of crushed pellet litter under broiler rearing conditions and an analysis of the operational costs of using WS additives are required for implementing this study.

## 1. Introduction

Bedding material intended for keeping birds should meet high-level and diverse requirements. A proper bedding material should act similar to a crepe paper, absorbing moisture and providing a dry, comfortable surface where the birds can bathe and rest. It should, therefore, be of low thermal conductivity, remain warm and act as insulation, be able to dry quickly, and should be soft, compressible, absorbent, and elastic. Since birds eat small amounts of the bedding material, it should be free from any impurities, chemicals, pathogens, and moulds, which may harm the health of birds. The bedding material should not be sticky or cakey, nor it should form layers [1].

Traditional bedding materials in the form of wood chips and sawdust are becoming increasingly expensive owing to a high demand for the same from the energy sector; hence, alternative bedding materials are being actively sought, one of which is straw. However, as the straw litter contains particles with sharp edges, which poses harm to the health and safety of the birds, the straw is pelleted after milling and the pellets produced are crushed. Bedding in the form of pelleted straw has favourable mechanical, chemical, sorption, and ecological properties [2]. The water-holding capacity of wheat straw (WS) is greater than that of wood chips and rice husks, and the evaporation rate of WS is low [3]. Straw palletisation allows for the reduction in the proportion of dust, and the proportion of very fine particles in the raw material mixture can be reduced via pretreatment or addition of binding additives [4]. 

The material compaction process should comply with the requirements of the pellets intended for bedding. Furthermore, a vast majority of available research results are related to energy pellets.

The quality of the pellets depends mainly on the properties of the lignocellulosic material and moisture. The raw material moisture content (*MC*) is one of the most important factors influencing the bulk density and mechanical durability of the pellets during their storage [5]. If the *MC* is less than 8–12%, the high frictional forces in the compression area make pelleting difficult. When pelleting fibrous organic materials, the *MC* can be 16–28%; however, the best results have been achieved with an *MC* of 20–24% [6]. 

Other requirements are placed on the storage of the products. When the product is stored for 4 months, the *MC* of the pellets should be 11–13% [7]. Moist energy pellets require additional heat to evaporate the water, resulting in a reduction in heat output. As biomass is a natural hygroscopic material, it is important to provide watertight storage facilities. The fulfilment of the storage requirement is indispensable; however, the absorption of moisture from the surrounding environment is a desirable feature for bedding pellets. In general, to obtain pellets of an acceptable quality, it is important to choose the appropriate biomass, grinding and conditioning method, type of additives, and pressure agglomeration process [8].

Choosing a suitable temperature for the biomass type is important in the pelleting operation, as heat activates the natural bonds and lubricants in the biomass, especially when combined with optimal *MC*, which facilitates heat transfer and thermal softening [9]. Water acts as a binding agent to adjacent particles through Van der Walls forces, increasing the contact surface of the bonded particles [10]. In combination with temperature, the moisture in biomass facilitates the gelatinisation of the starch, protein denaturation, and solubility processes during the fibre extrusion [11], thereby improving the quality of the densification process [12].

The pellets produced from a blend of agricultural biomass using additives can compete favourably with wood pellets [13]. The addition of straw to wood biomass could provide a direction for the development of pellet production. With an increase in the proportion of straw in the blends, the specific compaction work, pellet density, and heat of combustion decreases [14]. For bedding pellets, these properties could be advantageous for both energy-related reasons in the pelleting process and acceptable water absorption by lower density pellets. Pelleting properties have also been improved by blending WS with rice straw [15].

Among the different materials and technical parameters of the pelleting process, the shape and characteristics of the particle size distribution and the geometric mean particle size of the mixture are important features. The durability and strength of pellets made of smaller particle sizes is greater than that of larger particles. In a unit volume of a mixture containing small-sized particles, the total contact area is large, facilitating the formation of bonds between the particles [16].

To ensure an acceptable durability of the pellets, the particle size of the compacted material should be less than 5 mm [17]; however, such a generalisation could be risky. For WS with an *MC* of 10% wb, the proper particle size was 1–3 mm [18]. Multiple particle sizes in the mixture improves the durability and strength of the pellets; however, an excessive amount of fine particles, with a size less than 0.5 mm, negatively influences the friction and quality of the pellets [18]. The proportion of fine particles in the mixture should not exceed 10–20%, as the quality of the feed pellets decreases and the fine particles increase the friction against the die [19] if this amount is exceeded. The addition of hay to straw with longer particles increased the homogeneity of the blend and pellet strength [20], as the longer and wider straw particles wrap the small particles and form a “solid bridge” [21]. The compaction pressure and particle size significantly influence the density of the logging residues [22], corn, and switchgrass barley straw pellets; however, the different particle sizes of the WS did not have a significant influence on the pellet density [23].

The cited data demonstrate that there is a lack of knowledge regarding the influence of the separated particle size of the WS fraction, in combination with additives, on the quality of the pellets. The research results presented in this article are focused on a blend of WS with calcium carbonate (CC) and cassava starch (CS) aimed at improving the physical, mechanical, and sorption properties of pellets intended for litter.

The aim of the study was to (1) assess the process of pressure agglomeration of milled pure WS and WS blends with additives of CS or CC from separated fractions with different particle sizes; (2) develop the characteristics and parameters of the pelleting process in addition to the single pellet density *ρ_p_*; (3) determine the pellet strength parameters subjected to compressive load; (4) determine the water absorption coefficients for uncrushed *k_u_* and crushed pellets *k*; and (5) determine the pelleting conditions of the mixtures and fraction particle sizes in terms of the criteria for obtaining a high pellet strength with the least and greatest water absorption by the uncrushed and crushed pellets, respectively.

The scientific novelty of this article is that the original research methods and valuable research results are presented for the first time. The essence of the novelty is the pressure agglomeration of pure WS and blends with CS or CC of seven fractions with different particle sizes. This is based on the concept that it is possible to indicate the pelleting conditions such that the produced pellets are not only strong but also characterised by low humidity absorption from air during storage and high water absorption in the crushed form as bedding.

So far, studies have been performed using materials of different moisture and temperature, die height, additive ratio [24], and a single densified dose [25]. The present study complements the scope of the research; however, it is based on the knowledge, experience, and conclusions from previous research results.

## 2. Materials and Methods

### 2.1. Material and Additives

The materials were pure WS and blends of WS with CS or CC additives with a ratio of 6% wt./wt. The *MC* was 30% wb. The additive ratio and *MC* were selected experimentally, in the range of 0–10% wt./wt. and 10–30% wb, respectively [24].

The milled WS was obtained from litter pellet producer Lootor Ltd. (Słońsk, Poland). The cut WS was milled in a hammer mill at 1488 rpm, equipped with 160 beaters and a perforated screen with holes of 8 mm in diameter. CC and CS, 2 kg each, were purchased in a hypermarket. The CC purity was 98.0%. The *MC* and mean geometrical particles size of the CC and CS were 6.7% and 6.8% wb, and 0.15 and 0.29 mm, respectively.

The *MC* was determined in accordance with the requirements of the S358.2 standard. Samples weighing 25 g were weighed on an electronic scale (WPS 600/C, Radwag, Radom, Poland) with an accuracy of 0.01 g and dried at 105 ± 2 °C for 24 h in a SLW 115 laboratory dryer (Pol-Eko Aparatura, Wodzisław Śląski, Poland). Measurements were made three times for each combination of factors.

### 2.2. Preparation of Samples

Demineralised water calculated on the basis of the relationship $m_{H_{2}O}=m_{1}\left(1-MC_{1}\right)/\left(1-MC_{2}\right)$ was added to the WS with an equilibrium moisture of approximately 7.5% wb; where *m*_H2O_ is the weight of the refill water, g; *m*_1_ is the initial WS mass, g; *MC*_1_ is the initial WS moisture; and *MC*_2_ is the desired WS moisture. The WS with an *MC* of 30% wb was stored in a climate chamber (3001-01, ILKA, Feutron, Greiz, Germany) and blended twice daily. After 1 week, the material was divided into three parts. One part was pure WS. CC or CS with a ratio of 6% wt/wt was added to the following parts. To two tanks with WS of 5 kg each a CC or CS with a weight of 0.319 kg each were added. Each blend was blended in a feed mixer (M-100, Metal Works, Przysiek, Poland) for 15 min, based on previous experience. The *MC* was tested and water was added until the *MC* was 30% wb. The material was stabilised in a climate chamber at 4 ± 1 °C and a relative air humidity of 65 ± 2% for 5 days, with one daily blending.

### 2.3. Separation of Fractions with Different Particle Sizes

After stabilisation of the moisture, milled WS and WS blends with the CC and CS additives were separated on a vibrating sieve separator LAB-11-200/UP (Eko-Lab, Brzesko, Poland) in accordance with the ANSI/ASAE S319.4 standard. After the experimental selection of the sieve set and separation time, the material was divided into seven fractions with geometric mean particle sizes of 0.21, 0.50, 0.71, 1.00, 1.37, 1.94, and 2.81 mm. One-time samples of the material, weighing 25 g, were screened for 600 s. Thirty tests were performed such that the mass of the fraction was at least 70 g. After separation, the separated fractions were blended, the moisture was verified, and demineralised water was added. After blending, the material was stored in a climate chamber. The material for testing was collected as required, controlling the moisture (in the case of a decrease in moisture by 0.2% wb, water was added).

### 2.4. Pelleting Process

The pelleting tests were performed under laboratory conditions (to reduce any random error) in an open die chamber, accurately reflecting the actual conditions. The material was pelleted in an open-chamber die. A chrome steel die with an outer diameter, an inner diameter, and a height of 60, 8, and 300 mm, respectively, was surrounded by two heating elements with a total power of 1.3 kW. The material temperature at a height of 15, 30, and 45 mm from the base of the die was verified with J-type thermocouples with an accuracy of 0.1 °C and was controlled by a 3710 ESM regulator [24]. The die was installed on the support of a universal testing machine, TIRAtest, and the compaction process was controlled by the Matest program (Matest, Łódź, Poland). The compaction speed was 10 mm·min^−1^, and the force was measured with an accuracy of 10 N in a range up to 10,000 N. The force-shift relations were recorded with a step of 0.03 mm. Based on the preliminary tests, the compacted material was heated to 78 °C [24]. This was achieved by heating the elements to 116 °C for an hour before pelleting. The length of the agglomerate, *l_a_*, filling the die open end was 80 mm.

In the initial phase of the test, the material was poured into the die opening (the die was closed at one end) till a height of 80 mm, measured from the base, was reached. Then, 0.1 g doses, weighed on an analytical balance WPA 40/160/C/1 (Radwag, Radom, Poland), with an accuracy of ± 0.1 mg, were poured into the die hole and compacted until the limit sensor was turned on at a height of 80 mm from the die base. The material was compacted along a distance *s_m_* until the maximum pressure was achieved (Figure 1).

The formed pellets were shifted by a piston along section *v_p_*. The total piston displacement was *s_e_* = *s_m_* + *v_p_*, where *s_m_* is the piston displacement at the maximum pressure in millimetre and *v_p_* is the pellet shift in the die in millimetre. The counter-pressure required to achieve the maximum pelleting force developed along the length of the opening (die height) was *l_c_* = *v_p_* + *l_a_*, where *v_p_* is the formed pellet shift in the die in millimetre and *l_a_* is the length of the agglomerate filling the die opening in millimetre. Repeatable maximum pressure values were achieved after compacting several dozen doses of material. For each measurement system under these stable conditions, at least six pellets, approximately 20 mm long, were formed.

### 2.5. Compaction Process and Pellets Parameters

For the individual pellets, the diameter *d*, length *l_p_*, mass *m_p_*, and density *ρ_p_* were determined. The pelleting process was characterised by the following parameters: piston displacement to the maximum pressure *s_m_*, pellet shift in the die *v_p_*, specific compaction work *L_s_*, and specific work of pellet shift in the die *L_v_*. The parameters *L_s_*, *L_v_*, and *ρ_p_* were related to the DM. The pellet density was determined 1 week after the pellet formation, when the pellet expansion was complete.

The equations for calculating the compaction parameters are summarised in Appendix A.

### 2.6. Pellets Strength to Compressive Load

Pellet compression tests were performed on a universal testing machine. A single pellet was placed on a steel plate and pressed in the radial direction with a 20 mm × 50 mm punch at a speed of 5 mm·min^−1^ until the piston moved 4 mm in the radial direction (with a pellet diameter of 8 mm). From the recorded force-displacement data, the specific pellet compression work *E_j_*, at which the pellet cracked, the elasticity modulus *E*, and maximum tensile strength *σ_c_* were calculated using the trapezoidal method [26]. Each pellet production system was measured thrice. The equations for calculating the compressive strength parameters of the pellets are summarised in Appendix A.

### 2.7. Water Absorption by Uncrushed and Crushed Pellets

All pellet samples were stored under the same ambient conditions; temperature of 22 ± 2 °C, relative air humidity of 65 ± 3%, and air pressure of 1012 ± 11 hPa. Under these conditions, the pellet material achieved an equilibrium moisture of approximately 7% wb.

Three uncrushed and crushed pellets were selected for each measurement system. The pellets were placed in mesh coffee infusers and weighed on an electronic laboratory balance (WLC1/A2, Radwag, Radom, Poland) with an accuracy of ± 0.01 g. The samples were immersed in demineralised water at a temperature of 20 ± 1 °C for 30 min. After removal from the water, the pellets in the infusers were allowed to drain for 5 min, and gently shaken three times to remove excess water from the sieve screen and weighed again [24].

The described method of water absorption testing was selected based on an analysis of the methods presented to date in the available literature [27].

The water absorption was calculated based on the weight ratio of the water absorbed by the pellets and the initial weight of the pellets related to the DM (Appendix A).

### 2.8. Statistical Analysis

The data was analysed in terms of the influence of the factors (material type and fraction particle size) on the parameters characterising the pressure agglomeration process (specific compaction work, specific work of pellet shift in the die, piston displacement, pellet shift in the die, and single pellet density), pellet strength (specific work compression, elasticity modulus, and tensile strength), and water absorption by the uncrushed and crushed pellets based on a Multivariate Analysis of Variance (MANOVA), using the F test (Fisher–Snedecor). The statistical significance of the differences between the mean values of the parameters was determined using the Tukey test method. The analyses were performed using the assumed significance level of *p* ≤ 0.05 using Statistica v. 13.3.

## 3. Results and Discussion

### 3.1. Pressure Agglomeration Process

The pressure agglomeration process was analysed based on the example of the compaction of pure WS with a particle size of 0.50 mm. To begin, a material sample (p1, Figure 1) was compacted; the pressure increase curve vs. piston displacement was approximately a straight line. The piston displacement during the sample compaction was 4.8 mm. During further piston displacement, the pellets were moved in the die opening until the maximum pressure at the end of the piston stroke was met, which was determined by the limit sensor.

Successively compacted pellets, located in the die opening, created an increasing resistance as a result of the back pressure. By the 14th sample (p14, Figure 1), the material compaction curve indicated a typical exponential course [18], and the pellet sliding resistance in the die was rather uniform. For subsequent samples, the back pressure systematically increased and the pellet movement began after achieving the maximum pressure, which dropped immediately.

The generated maximum compaction pressure resulted from the requirement to overcome the static friction between the peripheral surfaces of the pellet and the die opening. The agglomerate material particles at rest under the compacted sample formed temporary adhesive bonds with the surface of the die channel, which had to be broken. The moment the pellet started to move, the static friction changed to kinetic friction of a lower value. During the movement of the pellet, the back pressure was gradually reduced owing to the reduction in the total actual contact surface between the top of the material particles with the surface of the die opening. These types of frictional phenomena are characteristic of low movement speeds [28].

The developed external friction directly increases the back pressure and indirectly, the die temperature. The heat from the hot die diffuses directly to the nearest material particles and because of propagation enhanced by particle moisture (30% wb), it spreads radially in the agglomerate. The diffusion gradient depends on the type of material and water content in the material, which reduces the glass transition temperature limit [29]. Therefore, there is an enhancement in the effects favouring the developed friction and transformations of lignocellulosic compounds, which are thermoplastic polymers. Lignin and hemicellulose, under water saturation and high temperature conditions, change the elasticity modulus and reduce the viscosity of the polymers, changing from glass to rubbery, facilitating particle flow [18]. At a moisture of 30% wb, the biomass glass transition can start at a temperature of 24 °C [30].

The described characteristics of pure WS pelleting also apply to WS blends with CS or CC additives densified for different *x*; however, these factors influence the parameters of pressure agglomeration and pellets as well as water absorption by the uncrushed and crushed pellets.

### 3.2. Pressure Agglomeration Process and Pellet Parameters

Statistically, the pressure agglomeration parameters (specific compaction work and shift pellet, piston displacement, and pellet shifting in the die) were significantly (*p* < 0.0001) dependent on the type of additive and the *x* (Table 1).

Characteristic pellet parameters such as diameter, density, tensile strength, and water absorption by the uncrushed pellets differed significantly (*p* < 0.0001) in a statistical sense, with respect to the type of additive and *x* (Table 1 and Table 2).

The type of additive also influenced a statistically significant differentiation of the elasticity modulus, specific pellet compression work, and the water absorption coefficient by the crushed pellets. The *x* was insignificant against the specific pellet compression work (*p* = 0.1470) and water absorption by the crushed pellets (*p* = 0.1495). For the elasticity modulus, the *p*-value was at the limit of significance (*p* = 0.0482).

#### 3.2.1. Pellet Diameter d, Piston Displacement to the Maximum Compaction Pressure s_m_, Pellet Shift in the Die v_p_, Specific Pellet Compaction Work L_s_, and Specific Work of Pellet Shift in the Die L_v_

With a designed die opening diameter of 8.0 mm, the pellet diameter was 1.12–2.37% larger. This indicates the expansion of the pellets after they left the die and during their weekly storage. Pellets formed from pure WS had the largest diameter (8.19 mm); the pellets with additives had a smaller diameter of 8.11 mm (Table 1). The small difference between the pellet diameter and die opening allows us to suppose that the bonding stability between the particles with additives was greater than that for pure WS, which may influence the density and pellet strength.

With an increase in the densified *x*, from 0.21 to 1.00 mm, the diameter of the pellets increased from 8.09 to 8.19 mm, respectively. In the *x* range of 1.00–2.81 mm, the pellet diameter decreased from 8.19 to 8.14 mm. In the range of smaller sizes (0.21–0.71 mm), the pellets’ diameter changed marginally to 8.09–8.11 mm. The pellets formed from mixtures with a smaller *x* underwent less expansion and had greater tensile strength *σ_c_* than the pellets made from a larger *x*. The proof is the average, yet negative, correlation between the diameter and *σ_c_*, elasticity modulus for pellet compression *E* (Appendix A), and the surface diagram (Figure 2).

The pellets with the smallest diameter and high *E* were characterised by the highest *σ_c_*. The piston displacement at the maximum force/pressure *s_m_* was greater during the compaction of the blends than during that of the pure WS (Table 1). The compaction of blends with an increasingly larger *x* occurred at a greater *s_m_*. The greater the *s_m_*, the greater the change in material height during the applied pressure, the greater the material’s compaction ability, and greater specific compaction work *L_s_* were performed, because *L_s_* was correlated with the *s_m_* (r = 0.495). The relationship among the *s_m_*, *L_s_*, and *x* clearly indicates a significant increase in the *s_m_* at an *x* of 1.94 mm, and then a plateau is attained (Figure 3).

The pellet shift in the die *v_p_* was inversely correlated with the *s_m_* (r = −0.532). The greater the *v_p_*, the greater the *L_v_* required to be performed. The proof of the coherence between the variables is given by the extremely high correlation (r = 0.705). The relationships among these three variables are presented graphically (Appendix A). The inverse correlation between *v_p_* and *s_m_* is logical, as with a greater *s_m_*, the formed pellet had a reduced height and the *v_p_* corresponded to a shorter distance. Considering that the mass of the compacted dose was the same, 0.1 g, the low pellet height should translate into a higher density. Such a correlation was not found, although a positive tendency was noticed.

With a larger *x*, it was necessary to perform a greater *L_s_* (high correlation, r = 0.565). As there was also a positive correlation between the *L_s_* and *s_m_* (r = 0.495), synergy and a significant increase in the *L_s_* occurred in combination with *x* (Appendix A). The additives significantly increased the *L_s_*. During the compaction of the pure WS, the *L_s_* was 49.0 kJ·kg^−1^; the addition of CS and CC increased this to 64.7 and 72.2 kJ·kg^−1^, respectively, i.e., by 32% and 47%, respectively. The values of the *L_s_* were considerably high as the die height of 80 mm was conducive to the creation of a relatively high counter-pressure in the die chamber. Similar values of *L_s_* (47.1 kJ·kg^−1^) were achieved at 2% wt./wt. of CC or CS additives, and at a higher dose of 0.15 g, *L_s_* = 67.2 kJ·kg^−1^ [25]. Hop cones with 10% moisture content, compacted under the pressure of 40–80 MPa, required lower specific compression energy (14.20–24.48 kJ·kg^−1^) [31]. For energy-related reasons, it would be beneficial if the *L_s_* was less; however, the greater compaction work performed indicates a greater back pressure to be overcome. This could translate into an increase in the *ρ_p_*. The back pressure created in the die chamber was the effect of the external friction and transverse pressure, which was released owing to pellet expansion and axial pressure. The axial and transverse pressure distribution has been studied during miscanthus and switchgrass briquetting [32].

Despite the inverse correlation between the *s_m_* and *v_p_*, such consistency was not found between *L_s_* and *L_v_*. Conversely, when comparing the value relations of these works, it can be concluded that it was more effective to compact the blends than the pure WS and material with a larger, rather than smaller, *x*. The *L_s_* depended largely on the compaction pressure gradient in relation to the piston displacement, and the *L_v_* depended to a greater extent on the back pressure than on the path along which the pellets were moved in the die. Such a favourable relationship does not always occur. During the densification of the walnut shell, with a dose of 0.4 g, the *L_v_* was greater than the *L_s_* because the resulting agglomerates from a single dose were long and had to be moved over a longer distance in the die [33]. Walnut shells were not as susceptible to plastic deformation as WS.

The *ρ_p_* was the highest during the compaction of the WS blend with CC and amounted to 746 kg·m^−3^; the pellets from the WS and CS blend had only a marginally lower density (737 kg·m^−3^). The lowest density was found in the pure WS pellets (663 kg·m^−3^); the difference in relation to the blends was greater than 10%.

In the *x* ranged from 0.21 to 1.00 mm, the *ρ_p_* decreased from 798 to 644 kg·m^−3^, i.e., 19%. The compaction of the material with an *x* of 1.00–2.81 mm increased the *ρ_p_* from 644 to 796 kg·m^−3^, i.e., by 24%. The produced material pellets with a smaller *x* were characterized by a greater standard deviation (SD) than the pellets formed from fractions with a larger *x*. Compared to the smaller *x*, there was a greater elastic effect using the larger *x* of the material and greater porosity of the sample, which increased the transverse pressure, contributing to an increase in external friction. During the piston pressure on the material, the elastic effect in the axial direction was limited; however, it influenced the pressure propagation, which started from the piston side and decreased with the height of the agglomerate. The displacement and deformation of small, irregularly spherical or oval-shaped particles were smaller than those of the larger particles, which were in the shape of an irregular cylinder or cuboid. Long particles are easier to bend and compress; however, the combined long particles increase the attenuation from the fibrous parts of the straw. With less contact between the adjacent particles, the potential density decreases [34]. There is less contact between spherical or oval particles than between oblong particles. The exceptions were the smallest particles, containing numerous mineral parts, which were characterised by a considerably greater *ρ_p_* than lignocellulosic particles. In the fraction with the smallest *x* of 0.21 mm, the relative proportion of mineral impurities was likely high; hence, the considerably greater *ρ_p_* compared to the next fraction with an *x* of 0.50 mm. The *ρ_p_* made from each of these two fractions was 798 and 674 kg·m^−3^, respectively.

In summary, it can be concluded that a greater *ρ_p_* was achieved after including additives, which improved the conditions for the formation of strong bonds between the material particles, mainly cohesive bonds that do not break during stress relaxation and elastic expansion of the pellets after leaving the die chamber [35]. These types of bonding mechanisms occur not only between lignocellulose particles but also between particles of pharmaceutical powders and feed [10]. Thus, the 6% wt./wt. of CS or CC additives to the WS with 30% wb absorbed water effectively from the surface of the WS particles because of diffusion, creating an adhesive film, and increasing the adhesion of the agglomerate particles to the die surface. More water could evaporate from the better adhering agglomerate film to the hot die surface, thus improving the conditions for increasing back pressure.

*Ρ_p_* is a sufficient condition to assess the quality of the pellets; however, the necessary condition is the durability or strength of the pellets to external loads. If the application states that the pellet strength is acceptable, it implies that the *ρ_p_* is correct. There is no value to study a *ρ_p_* that is easily crushed. Thus, there is no feedback, as greater *ρ_p_* is not evidence of stronger bonds between the particles [36]. No relationship has been found between the density and durability of the pellets or briquettes [37]. The high density of the agglomerate could be related to the high specific density of the raw material. Our experience indicates that walnut shells with high specific density, the milled particles of which were characterised by high sphericity, had poor bonds and the pellets were characterised by low strength [33] compared to the pellets made of fibrous biomass with low specific density, yet more durable bindings [20]. Therefore, in the next section, the strength parameters of pellets subjected to compressive load are analysed.

#### 3.2.2. Elasticity Modulus for Pellet Compression E, Specific Pellet Compression Work E_j_, Tensile Strength σ_c_, Water Absorption by Uncrushed k_u_, and Crushed Pellets k

The values of the strength parameters (elasticity modulus for pellet compression *E*, specific pellet compression work *E_j_*, and tensile strength *σ_c_*) clearly prove that the additives of CS or CC increased the resistance of the pellets to radial compressive loads (Table 2). The strength parameters were correlated with the *ρ_p_* (Appendix A).

The *E* for the pellets formed from blends of WS with CS or CC was 7.19 and 7.99 MPa, respectively, and was greater than the *E* of pure WS pellets by 146% and 174%, respectively. Pellets made of pure WS with a moisture of 30% wb were characterised by plasticity and easy crushing deformation, with a poor cracking effect. It is likely that wax and pure WS lignin with a *MC* of 30% wb was not sufficiently active to increase the bond strength and stabilise the structure [38]. The addition of additives to WS increased the elasticity of pellets, as part of the water from the particles surface was absorbed by the CS and CC powders, which have a high moisture absorption capacity. Similar moisture uptake was observed when wheat flour was added to torrefied pine [39]. Compared to the addition of CS, the pellets of the CC blend demonstrated a greater *E* owing to the lower porosity of the CC powder with smaller *x* than the CS.

The *E* for the pellets made of blends with different *x* did not change significantly. Only a marginally increasing tendency of the *E* with the dimensions of the particles could be observed.

The *E* was highly correlated with the other two strength parameters, *σ_c_* and *E_j_*, for which the correlation coefficients were 0.626 and 0.506, respectively. The correlation coefficient between the *E* and *ρ_p_* was 0.456 (average correlation).

Compared to *E*, the *E_j_* until the pellets cracked was greater for agglomerates formed from blends with CC and CS additives than for pure WS, amounting to 5.31, 4.15, and 3.42 mJ·mm^−2^, respectively (Table 2). Compared to pure WS, the pellets from the blends of WS with CS and CC were more resistant to cracking by 21% and 55%, respectively. At 78 °C, the CS grains gelatinised [40], swelled, and formed a viscous colloidal solution with water [41], which when cooled, resulted in a sticky paste, increasing the pellet strength to cracking under compressive load. The addition of CS to spruce and pine wood increased the pellets durability [36]. Conversely, when wheat flour was added to torrefied pine, lower durability, hardness, and energy density were observed [39]. By creating spontaneous clusters, CC demonstrated superior mechanical properties and the ability to combine with lignocellulosic particles, increasing bond strength and resistance to cracking [42].

Other than the extreme fractions, the pellets made of blends with different *x* were characterised by a similar *E_j_*, with a certain increasing tendency, i.e., the larger the particles, the greater the crack resistance of the pellets. The best results were achieved with the blend of WS and CC. This finding is confirmed in the literature. Adhesive bonds related to the surface wettability [43] influence the fracture surface, which depends on the particle size distribution [44]. A positive relationship has been found between the wettability and adhesion of wood [45]. With a small *x*, the pellet fracture surface is between adjacent particles; with elongated particles, the pellet fracture surfaces indicate breakage of the long particles, and the pellet strength depends largely on the strength of the biomass fibres [44].

The highest *σ_c_* were characteristic for pellets made of the WS with CC blend and amounted to 5.23 MPa (Table 2). These stresses were greater than those for pellets formed from the blend of WS with CS and pure WS, by 30% and 107%, respectively. In general, the pellets made from the fractions with a larger *x* had a greater *σ_c_*, although there were illogicalities for the smallest fractions, 0.21 and 0.71 mm, especially for the blend pellets.

The phenomena described in relation to the *E* and in particular, the *E_j_* and the *ρ_p_*, which also influenced the maximum pellet strength, expressed as tensile strength. Tensile strength represents the intensity of the internal surface forces occurring in a cross-section perpendicular to the direction of the applied compressive load. *σ_c_* correlates highly with the *E* (r = 0.626), very highly with the *E_j_* (r = 0.750), and with an independent parameter, *ρ_p_* (r = 0.661). There was synergy between these variables. The greater the *E* and *σ_c_*, the greater the *ρ_p_* (Figure 4).

It was previously reasoned (Figure 2) that there is an inverse relationship between the *σ_c_* and pellet diameter in relation to the *E*. *E_j_* is the most valuable and stable parameter among the three parameters characterising the compressive load strength of pellets. It is calculated from the area under the compaction curve to the maximum point where the pellet breaks. *E* and *σ_c_* are calculated for the ordinate point. The choice of a force in terms of elasticity encounters difficulties in identifying the point at which an elastic load continues to exist. Choosing the maximum point is not difficult; however, it is burdened with a random error of the phenomenon.

An important parameter characterising the quality of pellets intended for bedding is the water absorption by the pellets, both uncrushed *k_u_* and crushed *k*. Whole pellets should be characterised by the low absorption of humidity from the air to ensure that they do not undergo biological decomposition during storage, and pellets after crushing should be characterised by high water absorption. Such features were characteristic of the produced pellets. The data (Table 2) indicate that it was beneficial to produce pellets with additives, as the *k_u_* was 1.64–1.70 g H_2_O·g^−1^ DM and the *k* increased to 4.62–4.95 g H_2_O·g^−1^ DM, virtually a factor of three. The pellets blended with CC had marginally improved absorption parameters compared to CS. The *k_u_* of the pure WS was 2.32 g H_2_O·g^−1^ DM and was 39% greater than the pellets from blends; for pellets after crushing, the water absorption increased by only 0.7 times, amounting to 3.90 g H_2_O·g^−1^ DM, i.e., 18% less than the pellets crushed from blends. From the results analysis of these studies, a practical conclusion can be drawn that pellets should be made from blends of WS with CS or CC and the pellets should be crushed immediately before being used for bedding.

The greater the *x* from which the pellets were produced, the greater the *k_u_*. *k* did not depend statistically significantly on the pellet *x*, yet in correlation with *k_u_*, a certain positive tendency can be observed. The greater the *x* and lesser the *k_u_*, the greater the *k* (Figure 5).

The best *k* was achieved for the products made of blends in the *x* range of 1.00–1.94 mm. The *k* was negatively and poorly correlated with the *E_j_* (r = −0.278) and *ρ_p_* (r = −0.287). Because the strength parameters and *ρ*_p_ were characterised by mutually positive and high correlation coefficients; the graphical relationship for *k* vs. *E_j_* and *E* is displayed in Figure 6.

Pellets made of blends, characterised by an *E* of 8–11 MPa and requiring a greater *E_j_* to crush the pellets, were the best in terms of water absorption. Moisture absorption can reduce the disruptive force and have a hardening effect through hydration and alkaline cross-linking catalysis, partially strengthening the bonds [21].

The conclusion regarding the positive effect of the use of additives, especially CC, and a material *x* of 1.00–1.94 mm in the production of pellets intended for litter should be reinforced by the argument that the additive of CC can increase N mineralisation, promote the formation of NH_4_^+^ [46], and reduce the pH of the litter [47]. Pellets made of rice husk with additives of lignin and calcium hydroxide had acceptable mechanical durability, stable combustion, and absorbed minimal moisture [48].

Pellets made from the blend of WS and CC with a moisture of 30% wb at 78 °C in a die with a diameter of 8 mm and a height of 80 mm demonstrated acceptable density, *ρ_p_*, of approximately 800 kg·m^−3^, acceptable strength parameters (*E*, *E_j_*, and *σ_c_*), and the best water absorption in the crushed form, *k*, of approximately 5.5 g H_2_O·g^−^^1^ DM.

The obtained test results in laboratory conditions should be verified on a semitechnical scale and comparative experiments should be performed with the use of crushed pellets in separate cages with birds, e.g., broilers. Research aimed at a cost analysis of producing pellets from WS blends with additives is also recommended.

## 4. Conclusions

Milled pure WS and blends of WS with additives of CS or CC with a ratio of 6% wt./wt., *MC* of 30% wb, and material temperature of 78 °C were pelleted with a single dose of 0.1 g in an open chamber die of 8 mm in diameter and of 80 mm in height. The samples of the mixtures were divided into seven fractions with a geometric mean *x* in the range of 0.21–2.81 mm. The best pellets were formed from the WS blend with CC from the fraction with an *x* of 1.00–1.94 mm. These pellets were characterised by a high density of 800 kg·m^−3^ and resistance to compressive loads, assessed by an *E* of 10 MPa, *E_j_* of 6.5 mJ·mm^−2^, and *σ_c_* of 6 MPa. Owing to the sorption properties of CC, the pellets from this blend had the lowest water absorption coefficient of 1.5 g H_2_O·g^−1^ DM after crushing the highest—5.5 g H_2_O·g^−1^ DM. The advantage of using the CC additive with the WS was to obtain pellets of an acceptable density, strength, and water absorption when crushed. A better understanding of details of the bonds between adjacent straw particles activated by these additives, in combination with the moisture and temperature of the material, can be achieved by using a scanning electron microscopy. An investigation of crushed pellets as litter in broiler farming and an analysis of the economic effects of adding CC to WS are recommended.

## Figures and Tables

**Figure 1 materials-13-04623-f001:**
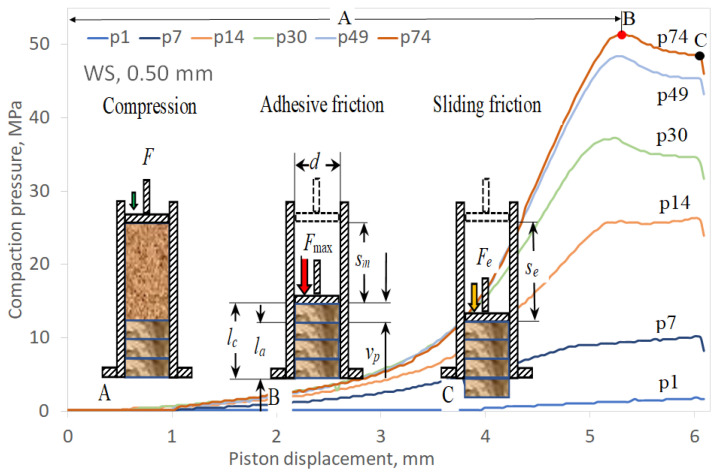
Schematic of sample pelleting cycle on background of selected curves of compaction pressure vs. piston displacement: (**A**) start of compaction of a new material dose, (**B**) attaining maximum compaction force/pressure, and (**C**) shift of currently compacted pellet along with agglomerate located under it. *d*—die (pellet) diameter, *l_a_*—agglomerate length, *l_c_*—real die height, *s_m_*—piston displacement at maximum agglomeration pressure, *s_e_*—end of piston displacement, *v_p_*—pellet shift in the die, *F*, *F*_max_, and *F_e_*—compaction forces: current, maximum, and pellet shift, respectively.

**Figure 2 materials-13-04623-f002:**
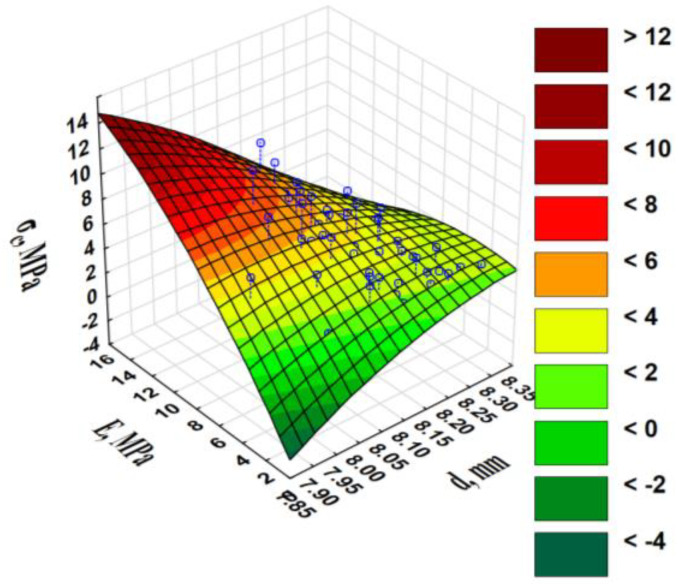
Pellet tensile strength *σ_c_* vs. pellet diameter *d* and elasticity modulus *E* during pellet compression.

**Figure 3 materials-13-04623-f003:**
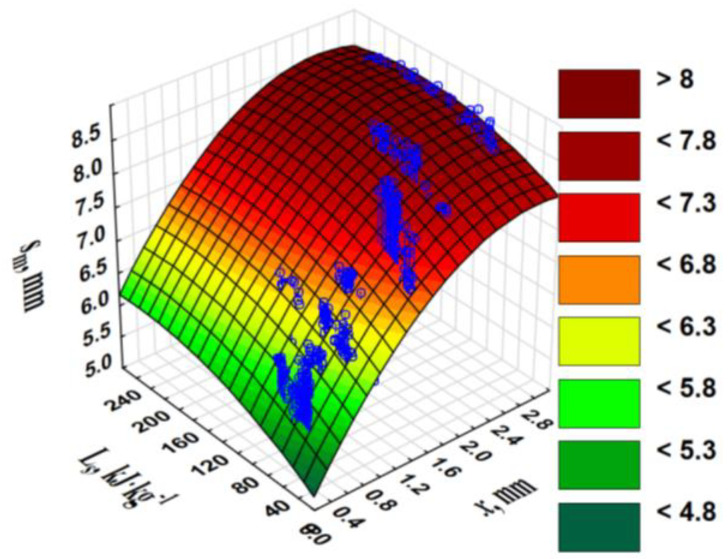
Piston displacement *s_m_* vs. fraction particle size *x* and specific compaction work *L_s_*.

**Figure 4 materials-13-04623-f004:**
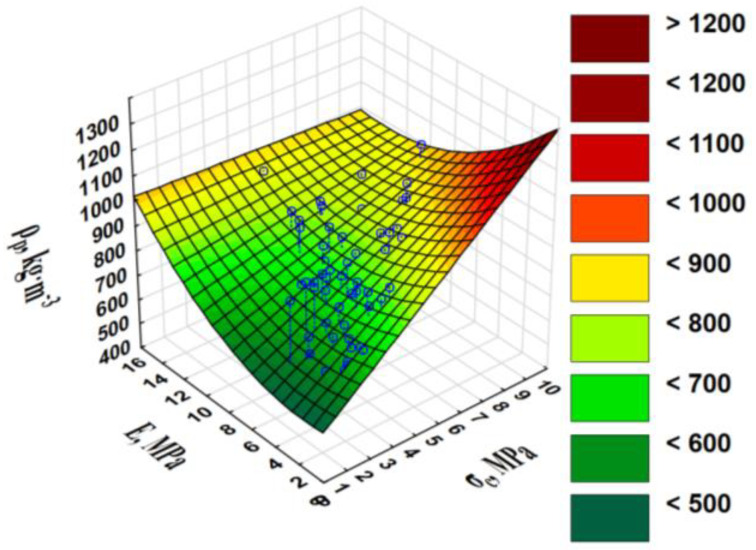
Single pellet density *ρ_p_* vs. pellet tensile strength *σ_c_* and elasticity modulus for pellet compression *E*.

**Figure 5 materials-13-04623-f005:**
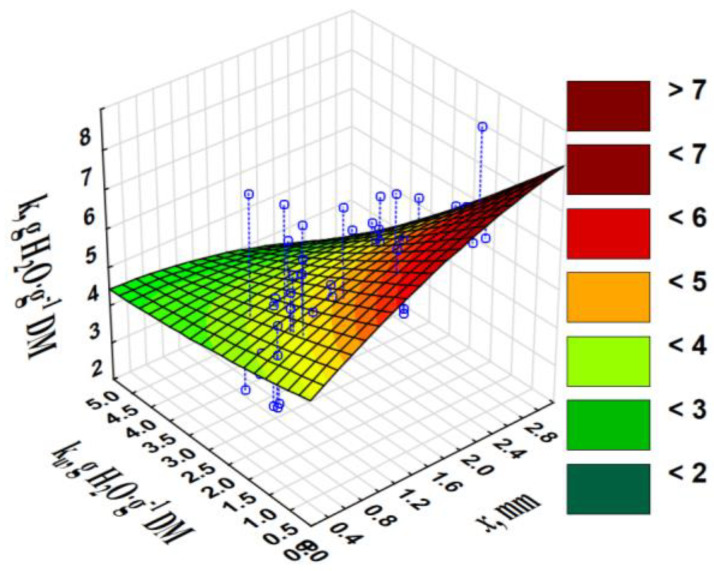
Water absorption by crushed pellets *k* vs. fraction particle size *x* and water absorption by uncrushed pellets *k_u_*.

**Figure 6 materials-13-04623-f006:**
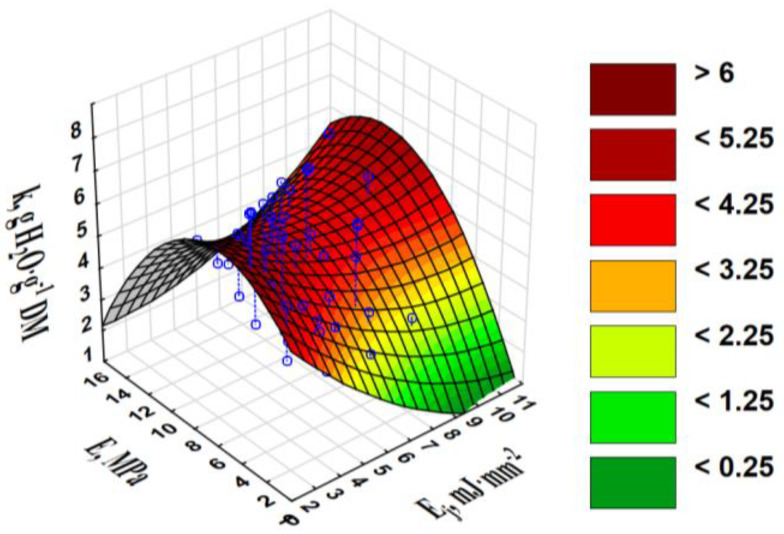
Water absorption by crushed pellets *k* vs. specific pellet compression work *E_j_* and elasticity modulus for pellet compression *E*.

**Table 1 materials-13-04623-t001:** Results of variance analyses, mean values (with three repetitions for each measurement setup), and SDs of pellet diameter, piston displacement at maximum agglomeration pressure, pellet shift in the die, specific compaction work, specific work for pellet shift in the die, and dry matter (DM) single pellet density for different types of additive and fraction particle size (*x*).

Factor	*d*, mm	*s_m_*, mm	*v_p_*, mm	*L_s_*, kJ·kg^−1^	*L_v_*, kJ·kg^−1^	*ρ_p_*, kg·m^−^^3^
*p*-value						
Additive	<0.0001	<0.0001	<0.0001	<0.0001	<0.0001	<0.0001
*x*	<0.0001	<0.0001	<0.0001	<0.0001	<0.0001	<0.0001
Mean and ±SD for type of additive						
NO	8.19 ^b^ * ± 0.07	6.38 ^a^ ± 0.86	0.68 ^c^ ± 0.25	49.0 ^a^ ± 22.9	22.3 ^c^ ± 8.4	663 ^a^ ± 82
CS	8.11 ^a^ ± 0.05	7.20 ^c^ ± 0.84	0.38 ^a^ ± 0.24	64.7 ^b^ ± 25.7	11.7 ^a^ ± 9.4	737 ^b^ ± 113
CC	8.11 ^a^ ± 0.08	6.74 ^b^ ± 0.93	0.54 ^b^ ± 0.30	72.2 ^c^ ± 49.6	15.2 ^b^ ± 7.9	746 ^b^ ± 137
Mean and ±SD for fraction, geometric mean of particle size on the *i*-th sieve *x_i_*						
0.21	8.09 ^a^ ± 0.09	5.91 ^b^ ± 0.26	0.61 ^b^ ± 0.25	41.9 ^a^ ± 12.4	15.87 ^bc^ ± 7.58	798 ^c^ ± 109
0.50	8.11 ^ab^ ± 0.07	5.59 ^a^ ± 0.37	0.61 ^b^ ± 0.29	48.3 ^b^ ± 13.0	20.08 ^de^ ± 10.03	674 ^ab^ ± 161
0.71	8.11 ^abc^ ± 0.07	6.29 ^c^ ± 0.24	0.58 ^b^ ± 0.27	55.8 ^c^ ± 26.0	19.42 ^de^ ± 8.06	650 ^a^ ± 138
1.00	8.19 ^d^ ± 0.10	6.37 ^c^ ± 0.38	0.63 ^b^ ± 0.30	49.6 ^b^ ± 13.8	18.60 ^cd^ ± 8.27	644 ^a^ ± 73
1.37	8.16 ^cd^ ± 0.05	7.52 ^d^ ± 0.52	0.59 ^b^ ± 0.32	37.6 ^a^ ± 10.4	10.86 ^a^ ± 5.86	726 ^b^ ± 89
1.94	8.14 ^bcd^ ± 0.06	7.93 ^e^ ± 0.25	0.38 ^a^ ± 0.25	83.9 ^d^ ± 27.8	12.06 ^ab^ ± 5.79	731 ^b^ ± 84
2.81	8.14 ^abcd^ ± 0.04	7.91 ^e^ ± 0.26	0.40 ^a^ ± 0.26	113.7 ^e^ ± 46.6	22.39 ^d^ ± 13.19	786 ^bc^ ± 45

*d*, pellet diameter; *s_m_*, piston displacement to maximum pressure; *v_p_*, pellet shift in die; *L_s_*, specific compaction work; *L_v_*, specific work for pellet shift in die; *ρ_p_*, DM single pellet density. NO, no additive. The values for NO and an additive ratio of 0% wt/wt are equivalent. * different letters in each column and factors within a value represent a significant difference at *p* < 0.05 using Tukey’s test.

**Table 2 materials-13-04623-t002:** Results of variance analyses, mean values (with three repetitions for each measurement setup), and SDs of the elasticity modulus for pellet compression, DM-specific pellet compression work, DM-specific work for pellet shift in die, tensile strength pellet compression, water absorption by uncrushed pellets, and water absorption by crushed pellets for different type of additive and fraction particle size (*x*).

Factor	*E*, MPa	*E_j_*, mJ·mm^−2^	*σ_c_*, MPa	*k_u_*, g H_2_O·g^−1^ DM	*k*, g H_2_O·g^−1^ DM
*p*-value					
Additive	<0.0001	<0.0001	<0.0001	<0.0001	0.0053
*x*	0.0482	0.1470	<0.0001	<0.0001	0.1495
Mean and ±SD for type of additive					
NO	2.92 ^a^ * ± 1.11	3.42 ^a^ ± 1.38	2.52 ^a^ ± 0.86	2.32 ^b^ ± 0.74	3.90 ^a^ ± 1.03
CS	7.19 ^b^ ± 1.73	4.15 ^a^ ± 1.27	4.03 ^b^ ± 1.17	1.70 ^a^ ± 0.38	4.62 ^bc^ ± 1.15
CC	7.99 ^b^ ± 2.71	5.31 ^b^ ± 2.17	5.23 ^c^ ± 2.15	1.64 ^a^ ± 0.56	4.95 ^c^ ± 1.17
Mean and ±SD for fraction. geometric mean of particle size on the *i*-th sieve *x_i_*					
0.21	6.74 ^a^ ± 2.28	4.18 ^a^ ± 1.97	4.78 ^c^ ± 2.36	1.43 ^a^ ± 0.28	3.60 ^a^ ± 0.77
0.50	5.28 ^a^ ± 1.48	3.33 ^a^ ± 1.18	2.37 ^a^ ± 0.95	1.65 ^a^ ± 0.38	5.00 ^a^ ± 1.52
0.71	6.33 ^a^ ± 3.28	3.93 ^a^ ± 2.03	4.26 ^bc^ ± 2.33	2.02 ^ab^ ± 0.32	4.46 ^a^ ± 1.16
1.00	5.14 ^a^ ± 2.95	4.72 ^a^ ± 1.67	3.04 ^ab^ ± 0.81	1.83 ^ab^ ± 0.42	4.49 ^a^ ± 1.41
1.37	6.85 ^a^ ± 4.28	4.74 ^a^ ± 2.46	3.95 ^bc^ ± 1.80	1.69 ^a^ ± 0.92	4.49 ^a^ ± 0.76
1.94	6.46 ^a^ ± 3.43	5.19 ^a^ ± 1.49	4.75 ^c^ ± 1.79	2.28 ^b^ ± 0.43	4.70 ^a^ ± 1.14
2.81	5.45 ^a^ ± 2.44	3.97 ^a^ ± 1.35	4.35 ^c^ ± 1.22	2.29 ^b^ ± 1.00	4.70 ^a^ ± 1.22

*E*, elasticity modulus for pellet compression; *E_j_*, specific pellet compression work; *σ_c_*, pellet tensile strength; *k_u_*, water absorption by uncrushed pellets; *k*, water absorption by crushed pellets. NO, no additive. The values for NO and an additive ratio of 0% wt/wt are equivalent. * different letters in each column and factors within a value represent a significant difference at *p* < 0.05 using Tukey’s test.

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
