# Peer review of "Influence of Fraction Particle Size of Pure Straw and Blends of Straw with Calcium Carbonate or Cassava Starch on Pelletising Process and Pellet"

_materials, 2020, doi:10.3390/ma13204623_

Round 1
Reviewer 1 Report
The introduction provides a good background of the topic. The manuscript is well structured and organized but the resolution of the figures is too low. The quality of the figures must be improved.

Author Response
Reviewer 1
Comments and Suggestions for Authors
The introduction provides a good background of the topic. The manuscript is well structured and organized but the resolution of the figures is too low. The quality of the figures must be improved.
We would like to thank the Reviewer very much for reviewing the article and for all the comments and suggestions that contributed to the improvement of the paper. We really appreciate the hard work you have done and want to acknowledge the tremendous contribution. Thank you very much. We improved all Figures by increasing the font.
Reviewer 2 Report
The manuscript, “Influence of Fraction Particle Size of Pure Straw and Blends of Straw with Calcium Carbonate or Cassava Starch on Pelletizing Process and Pellet (materials-962787)” describes a study to obtain pellets of optimized density, strength, and water absorption when crushed. These pellets were made using wheat straw and blends of wheat straw and calcium carbonate (CC) or cassava straw (CS) of different ratios.
The work is important. Authors discussed the originality of this work and provided the detailed experimental designs, observations and results. Authors reported density, tensile strengths, elasticity modulus of various pellets and used statistical method to determine the optimized properties of pellets.
However, the paper might not be accepted in its current version as it did not discuss the chemical effects of calcium carbonate or cassava starch at all. It would have been better if authors had provided SEM (scanning electron microscope) images of these pellets. Thus, authors are requested to provide those details in order to enhance the overall quality of the paper.
Author Response
We would like to thank the Reviewer very much for reviewing the article and for all the comments and suggestions.
Answer: Comment on SEM is important. We have experience in this method. This article is already extensive. It contains about 8,000 words and adding a description of this method would entail additional information in three chapters: Introduction, Method, and Results. It is impossible to present the entire spectrum of research in one article, especially with the journal's publishing limitations. Thank you very much for your suggestion and we will include it in future studies. We have added a sentence to the Conclusions to indicate the need for such research (Lines 555-558).
A better understanding of details of the bonds between adjacent straw particles activated by these additives, in combination with the moisture and temperature of the material, can be achieved by using a scanning electron microscopy.
Reviewer 3 Report
Manuscript Number: Materials-962787
Title:” Influence of Fraction Particle Size of Pure Straw and Blends of Straw with Calcium Carbonate or Cassava Starch on Pelletising Process and Pellet”
The paper describes the pressure agglomeration process of wheat straw and the blends of wheat straw with calcium carbonate or cassava straw with a ratio of 6% wt./wt. from seven separate fractions with sizes in the range 0.21–2.81 mm.
The topic is interesting although pelletizing process has been well studied and results were available in several papers.
The paper requires major revision and improvement in terms of proper citation.
KEYWORDS
- Keywords should not be repeated in the title. Please, verify and correct.
INTRODUCTION
- L152-153. The following sentence ”The pelleting tests were performed under laboratory conditions (to reduce any random error) in an open die chamber, accurately reflecting the actual conditions” should be moved in Materials and Methods Section
MATERIALS AND METHODS
- I suggest the authors to improve the paper by adding a section concerning the durability test procedure. The following references could be useful to reach this objective:
- Pampuro, P. Busato, E. Cavallo, Effect of Densification Conditions on Specific Energy Requirements and Physical Properties of Compacts Made from Hop Cone, Energies, 11 (2018) 2389.
- Kaliyan, R.V. Morey, Factors affecting strength and durability of densified biomass products, Biomass Bioenergy 33 (2009) 337-359.
- “The MC was 30% wb.” However, in the Introduction Section you affirmed “…the best results have been achieved with an MC of 20–24%”. Please, clarify.
- Section 2.4. In my opinion, the process you have presented is not a pelletizing process. To produce compacts you have described a kind of press and not a pelletizing machine: no pistons are present in this kind of equipment. The equipment you have described, is similar to the hydraulic press presented by Pampuro et al. (2018) in their work focused on hop cones densification. Please, clarify.
- Pampuro, P. Busato, E. Cavallo, Effect of Densification Conditions on Specific Energy Requirements and Physical Properties of Compacts Made from Hop Cone, Energies, 11 (2018) 2389.
- You have performed the experiment heating the material at 78°C. Please, explain the reason of your choice.
RESULTS AND DISCUSSION
- L301-302. Delete “transition”: there is a repetition.
TABLE CAPTIONS
- Authors reported mean values. How many replicates? Please, clarify.
Author Response
Reviewer 2
Comments and Suggestions for Authors
Manuscript Number: Materials-962787
Title:” Influence of Fraction Particle Size of Pure Straw and Blends of Straw with Calcium Carbonate or Cassava Starch on Pelletising Process and Pellet”
The paper describes the pressure agglomeration process of wheat straw and the blends of wheat straw with calcium carbonate or cassava straw with a ratio of 6% wt./wt. from seven separate fractions with sizes in the range 0.21–2.81 mm.
The topic is interesting although pelletizing process has been well studied and results were available in several papers.
The paper requires major revision and improvement in terms of proper citation.
We would like to thank the Reviewer very much for reviewing the article and for all the comments and suggestions that contributed to the improvement of the paper. We really appreciate the hard work you have done and want to acknowledge the tremendous contribution. We hope that we have responded to all comments and made all changes accordingly. Thank you very much.
Below are the answers to each comment. In the text, in order to facilitate the revision of the article, all changes were marked in blue.
KEYWORDS
- Keywords should not be repeated in the title. Please, verify and correct.
Answer (A): Thank you very much for this comment. You are right. I know that, but sometimes there is a lack of attention. Instead of the previous keywords: additive ratio; agglomeration; particle size; pelletising process; pellet strength; water absorption, I wrote down a new ones: biomass; additives; pressure agglomeration; pellet strength; water absorption.
INTRODUCTION
- L152-153. The following sentence ”The pelleting tests were performed under laboratory conditions (to reduce any random error) in an open die chamber, accurately reflecting the actual conditions” should be moved in Materials and Methods Section
A: So I did. I moved this sentence to the Ls 200-201 and removed the sentence "The material was pelleted in an open-chamber die." In order to not repeat the information. Our intention was to write in the Introduction why the research was conducted in an open die. At the end of the Introduction, it is usually advisable to write the aim of the work and briefly describe the reasons and conditions under which the research were carried out. It is important for the reader what to expect in the detailed descriptions.
MATERIALS AND METHODS
- I suggest the authors to improve the paper by adding a section concerning the durability test procedure. The following references could be useful to reach this objective:
- Pampuro, P. Busato, E. Cavallo, Effect of Densification Conditions on Specific Energy Requirements and Physical Properties of Compacts Made from Hop Cone, Energies, 11 (2018) 2389.
- Kaliyan, R.V. Morey, Factors affecting strength and durability of densified biomass products, Biomass Bioenergy 33 (2009) 337-359.
A: Thank you for this comment. It is important to assessing the quality of pellets, but those produced on a larger scale, as 0.5 kg of pellets is needed for one trial. When producing single pellets, there is no economic justification for durability test of pellets. Preparation of the fraction required 30 tests. Production of a single pellet under stable conditions required 76-98 compactions within 4 hours. 700 hours were spent on these tests. 6 pellets were produced for each measuring set. At least 3 pellets were intended for strength tests for each of the 7 fractions, two additives and pure wheat straw. For this combination, at least 3 pellets were tested for the water absorption capacity of the whole (uncompressed) pellets. After the strength test, the compressed pellets were also tested for their water absorption capacity.
Thank you for pointing out these references. I know both articles. Kaliyan and Morey's article is cited. The second article (Pampuro et al., 2018) is about a different material. The article is interesting and original, therefore I decided to cite it:
Ls381-383: Hop cones with 10% moisture content, compacted under the pressure of 40-80 MPa, required lower specific compression energy (14.20–24.48 kJ·kg–1) (Pampuro et al., 2018).
- “The MC was 30% wb.” However, in the Introduction Section you affirmed “…the best results have been achieved with an MC of 20–24%”. Please, clarify.
A: Thank you for this comment. I would like to clarify that in the Introduction (L99) I have given moisture values that do not include cassava or calcium carbonate additives. In L165, I stated that the selection of the proportion of additives and moisture was made on the basis of previous studies that we conducted and published in our manuscript. The research presented in this article is focused on material fractions, i.e. the particle size of crushed wheat straw with binding additives, with the key purpose of pellets for bedding.
- Section 2.4. In my opinion, the process you have presented is not a pelletizing process. To produce compacts you have described a kind of press and not a pelletizing machine: no pistons are present in this kind of equipment. The equipment you have described, is similar to the hydraulic press presented by Pampuro et al. (2018) in their work focused on hop cones densification. Please, clarify.
- Pampuro, P. Busato, E. Cavallo, Effect of Densification Conditions on Specific Energy Requirements and Physical Properties of Compacts Made from Hop Cone, Energies, 11 (2018) 2389.
A: Thank you for this comment. Indeed, under production conditions, pelletizing machines usually do not have a piston. The material sample is pressed into the die opening by means of a compacting roller. The pelleting theory and the pressure distribution under the roll in contact with the agglomerated material in the opening are described in the literature, although they are still not fully explained. It is possible to build a stand that would recreate such pelleting conditions. The same pressure effect on the agglomerated material can be achieved by using a piston. This technical solution is commonly used. Tests are carried out in a closed chamber more often. The die with an open chamber reproduces the real conditions better. This, of course, is very time-consuming. But the obtained results can be related to the conditions of the real die. The primary reference is the backpressure provided by the pellets in the die chamber, not the closed die bottom. For the tests, we use a universal testing machine with a mechanical drive (a toothed bar and a gear wheel driven by an electric motor). The hydraulic drive is less suitable for testing the pelleting process.
The reconstruction of the actual pelleting process also consists of the fact that a single dose of the material was 0.1 g. The dose was selected on the basis of the efficiency analysis of the actual production of pellets and measurements of the dividing planes visible on the produced pellets. Separate layers were weighed. In the range of this mass, studies were carried out with various doses. Final dose selection was based on the acceptable strength of the pellets. Therefore, pellets were made by compacting several dozen doses successively poured into the die chamber. Tablets are made of a single dose and the compaction is on both sides - to increase tablet durability and strength and provide a smoother tablet surface.
- You have performed the experiment heating the material at 78°C. Please, explain the reason of your choice.
A: Thank you for this comment. As I explained before, we conducted research in this field earlier. The beginning of the sentence explains this, but for the clarity, I refered to our manuscript at the end of the sentence [24]. (L210).
RESULTS AND DISCUSSION
- L301-302. Delete “transition”: there is a repetition.
A: Thank you for this comment. It is indeed a repetition. I have removed "the transition of". Thank you.
TABLE CAPTIONS
- Authors reported mean values. How many replicates? Please, clarify.
A: Than yous for this comment. I added in parentheses (with three repetitions for each measurement setup). However, I emphasize that all information on the number of repetitions were given in the Materials and methods chapter: L175, L246, L252.

Round 2
Reviewer 3 Report
Manuscript Number: Materials-962787
Title:” Influence of Fraction Particle Size of Pure Straw and Blends of Straw with Calcium Carbonate or Cassava Starch on Pelletising Process and Pellet”
The paper has been improved following my suggestions.
In my opinion, is now suitable for publication in present form.
Author Response
We would like to thank the Reviewer very much for recommending our article for publication.